# Association of Socioeconomic Status and Reasons for Companion Animal Relinquishment

**DOI:** 10.3390/ani14172549

**Published:** 2024-09-02

**Authors:** Sonya McDowall, Susan J. Hazel, M. Anne Hamilton-Bruce, Rwth Stuckey, Tiffani J. Howell

**Affiliations:** 1School of Psychology and Public Health, La Trobe University, Bundoora, VIC 3082, Australia; r.stuckey@latrobe.edu.au (R.S.); t.howell@latrobe.edu.au (T.J.H.); 2School of Animal and Veterinary Science, Roseworthy Campus, The University of Adelaide, Adelaide, SA 5005, Australia; susan.hazel@adelaide.edu.au; 3Adelaide Medical School, The University of Adelaide, Adelaide, SA 5055, Australia; anne.hamilton-bruce01@adelaide.edu.au

**Keywords:** pet–owner relationships, human–animal bond, socioeconomic populations, companion animal surrender, human health outcomes, Social Determinants of Health

## Abstract

**Simple Summary:**

Companion animals continue to be relinquished by their guardians to shelters. Historically, animal behavior has been seen as the main cause for companion animals to be relinquished. Therefore, we wanted to understand whether companion animals are relinquished due to issues related to the animal themselves or due to challenges in the companion animal’s guardian’s life. Previous research indicates that socioeconomic difficulties of the guardian, such as financial constraints and inadequate housing, contribute to companion animal relinquishment. Relinquishment data are crucial for a better understanding of how socioeconomic factors intersect with the care of companion animals. This study analyzed relinquishment data from five Australian shelters over the period of Australian Financial Years (FY July 1 to June 30) 2018/19 to 2022/23 and analyzed data on the 46,820 companion animals relinquished. The most common reason for relinquishment was housing for both low and high socioeconomic groups. Financial constraints were reported more in lower socioeconomic groups, whereas healthcare of the human as a reason for relinquishment was greater in the higher socioeconomic groups. This highlights the need for comprehensive, cross-disciplinary strategies that address both human and animal welfare needs holistically. Such integrated approaches can improve outcomes for companion animals and their human caregivers by addressing the underlying socioeconomic factors affecting their well-being.

**Abstract:**

It is important to understand the reasons for companion animal relinquishment to help reduce the financial and ethical problems arising from too many dogs and cats in shelters. This study investigates the socioeconomic factors and reasons behind companion animal relinquishment in Australia, utilizing data from five animal shelters, over a five-year period (Financial Year 2018/19 to 2022/23). Descriptive statistics reveal that the median Index of Relative Socio-Economic Advantage and Disadvantage (IRSAD) decile of companion animal guardians who relinquished their companion animal was decile 4 out of 10, indicating that they live in areas of lower-than-average socioeconomic status. Cats accounted for 59.4% and dogs for 40.6% of all relinquishments, with more relinquishments from lower socioeconomic deciles (1–5) (cats: 62.6%, dogs: 65.8%). The median age of relinquished cats was 5 months and dogs 16 months, with human factor-related issues (e.g., Housing, Financial Constraints, Human Healthcare) cited in 86% of cases. Descriptive analysis for the five financial years shows a declining trend in numbers of relinquishments, with housing issues (31.2%) identified as the primary reason, followed by ownership decisions (16.2%), financial constraints (11.2%), and human health issues (10.4%). Comparing the reasons for relinquishment between lower (decile 1–5) and higher (decile 6–10) socioeconomic status demonstrated that financial difficulties were a more common reason in lower vs. higher socioeconomic groups, while human health and family-related issues are more common in higher vs. lower socioeconomic groups. These findings highlight the critical role of socioeconomic factors in understanding why people relinquish their companion animals, which can inform targeted interventions to support companion animal welfare across different socioeconomic backgrounds.

## 1. Introduction

Social Determinants of Health (SDH) impact all aspects of our lives, and encompass the social, physical, and economic conditions in society that affect health [1]. These determinants include aspects of a person’s life, such as income, education, social support, access to healthcare, and environment, as well as the circumstances in which a person is born, grows up, lives, works, and ages [2,3]. However, research investigating the impacts of SDH on companion animal welfare is limited, despite the recognized important role of companion animals in their relationship with human guardians.

Improving SDH outcomes for humans could lead not only to better human but also animal welfare. Research has identified SDH reasons that lead to companion animal guardians relinquishing their companion animals, such as the inability to afford veterinary care or manage the animal’s health condition [4]. SDH factors affecting the animal’s guardian, such as limited economic resources, can restrict the animal’s access to appropriate training and veterinary services, which can lead to problem behaviors in dogs, negatively influencing human–animal interactions [5]. Poverty and inadequate housing can lead to limited access to required resources to provide care to companion animals, impacting negatively on their welfare, which may lead to neglect, which can then result in aggressive behaviors or poor physical body condition [4]. Aggressive or destructive behaviors can lead to relinquishment to a shelter, which could be a result of the companion animal guardian’s inability to understand the animal’s behaviors and needs [6,7].

The human and animal sectors have historically worked in silos, without truly integrating related public policy or service delivery to achieve positive outcomes for both humans and animals [4]. Integrating the SDH with the “Five Domains” model of animal welfare has enabled us to create a framework to identify how SDH in human guardians may influence animal welfare outcomes [8]. Relinquishment data serve as a critical indicator in this context, offering insights into the intersection of socioeconomic factors and companion animal care. The aim of this study is to identify the associations between an SDH proxy (i.e., socioeconomic status area, or SES area) and the reason for companion animal relinquishment. This underscores the broader implications of SDH on animal welfare, highlighting the need for integrated approaches that address both human and animal welfare needs synergistically. This research aims to bridge the gap between human and animal welfare paradigms by demonstrating the relevance of SDH in understanding and mitigating companion animal welfare issues, and animal relinquishment outcomes, thereby advocating for more comprehensive, cross-disciplinary strategies aimed at improving outcomes for both populations.

## 2. Materials and Methods

This project received approval from the La Trobe University Human Ethics Committee (HEC23190).

### 2.1. Organizations

Eight Australian rescue and shelter organizations were contacted via email and telephone to discuss if they maintained a database of owner-relinquished dogs and cats for the previous five financial years. A combination of larger and smaller organizations was contacted; generally, only larger organizations maintained the intake details required for this study. A variety of organization types in different locations were contacted for this study to enable as broad coverage of the Australian companion animal population as possible.

### 2.2. Social Economic Index Areas

The data from the shelters provided the postcode of the animal that was relinquished, which was used as the socioeconomic proxy measure of the companion animal guardian. We utilized the Australian Bureau of Statistics Socio-Economic Index for Areas (SEIFA) data, which are derived from the Census and divide Australian postcodes by social and economic characteristics [9]. The Census is conducted every five years and combines data such as income, employment, education, housing, occupation, and family structure to develop the SEIFA index. SEIFA consists of four indices used to describe the socioeconomic status of each postcode in that area: (1) Index of Relative Socioeconomic Disadvantage, (2) Index of Relative Socioeconomic Advantage and Disadvantage, (3) Index of Economic Resources, and (4) Index of Education and Occupation.

We used the Index of Relative Socioeconomic Advantage and Disadvantage (IRSAD) as the socioeconomic status tool, as it is recommended when comparing a range of areas [9]. The IRSAD score is a composite measure that combines various indicators of advantage and disadvantage, such as household income, internet accessibility, education, occupation, employment rate, property ownership, mortgage status, and health. This score is standardized across the nation, with a mean of 1000 and a standard deviation of 100. A location receives a score of 1000 if all indicators match the national average; otherwise, the score adjusts up or down based on how the indicators compare to the national average. The areas are arranged in ascending order based on their scores, ranging from the lowest to the highest. The bottom 10% of areas are labeled as decile 1, and this labeling continues incrementally up to decile 10, which represents the top 10% of areas. Higher IRSAD deciles, therefore, indicate greater socioeconomic advantage, while lower deciles signify more socioeconomic disadvantage.

### 2.3. Data Collection

Five organizations provided the researchers with existing organizational data that included the postcode of the animal’s residence prior to relinquishment, along with the animal species (i.e., dog or cat), age and sex of the animal, and the main reason for relinquishment, as provided by the person relinquishing the animal. The five organizations are based throughout Australia, including two from Victoria, and one organization each from Queensland, New South Wales, and South Australia. The data were extracted for five financial years (July 1 to June 30) to cover the time pre- and post-COVID-19, from Financial Years 2018/2019 to 2022/2023. No details of the person relinquishing the animal were obtained.

### 2.4. Data Analysis

Data were analyzed using IBM SPSS Statistics (version 29.0.2.0). Datapoints with missing postcode, relinquishment reason, and/or species were excluded from analysis. The total number of datapoints included in the analysis was *n* = 46,820 (Table 1).

All data were non-normally distributed. A categorical logistic regression to investigate relationships between relinquishment reasons, SES low/high deciles, and financial year was attempted, but a goodness-of-fit test showed a poor fit for the overall model. Therefore, we used a chi-square test to measure if there was any significant association between relinquishment reason and SES low/high deciles.

Adjusted Pearson residuals were used to identify the associations between relinquishment reason and SES low/high deciles. Adjusted Pearson residuals are provided in the results of chi-square tests and are functionally identical to z-scores for non-normal data. They represent data that are transformed to become normally distributed, such that absolute values greater than 1.96 will have a *p*-value of less than 0.05 (2 × 2 test) [10]. Absolute values greater than 2.33 have *p* < 0.01, and absolute values greater than 3.09 have *p* < 0.001.

Due to multiple organizations providing data, there were several variations as to reasons for relinquishment. All relinquishment data provided were inspected by S.M. and categorized to enable an analysis of nine relinquishment groups, which were discussed in depth and agreed to by all authors. A total of 250 coded relinquishment reasons were provided across the five organizations, which were then condensed into the nine relinquishment reason groups (Table 2). A full list of these codes can be found in Appendix A.

The relinquishment reason groups were also further reduced into two groups: human factors and animal factors (Table 2). Human factors were evaluated as reasons for relinquishment attributable to aspects related to humans, such as human health, whereas animal factors encompass reasons stemming from the animals themselves, such as aggression.

## 3. Results

### 3.1. Descriptive Statistics

The median IRSAD decile of companion animal guardians who relinquished their companion animal was decile 4 of a total of 10 deciles (1 lowest–10 highest). The total relinquishments consisted of 59.4% cats and 40.6% dogs, of which 62.6% of cats came from the low socioeconomic status (deciles 1–5) group and 37.4% from the high socioeconomic status group (decile 6–10) (*n* = 27,822). Dogs were similar, with 65.8% of relinquishments from low socioeconomic status (decile 1–5) and 34.2% from high socioeconomic status (decile 6–10) (*n* = 18,998). The median age of companion animals at relinquishment was 5 months for cats and 16 months for dogs. Human factor-related reasons accounted for 86% of relinquishments, with only 14% for animal factor-related reasons. There was a numerical decline over the last five (5) financial years for numbers of both dogs and cats relinquished (Table 3).

### 3.2. Socioeconomic Analysis

The most common overall reason for relinquishment was Housing (31.2%) followed by Ownership/Guardian decisions (16.2%), Financial Constraints (11.2%), and Health of Human (10.4%) (Table 4). The Pearson chi-square test comparing reasons for relinquishment by IRSAD decile was significant, χ^2^ = 501.42, *df* = 8, *p* < 0.001. The relinquishment rates of companion animals vary across Index of Relative Socio-Economic Advantage and Disadvantage (IRSAD) postcode deciles, showing a higher number of companion animals relinquished in deciles 1–5 compared to deciles 6–10 (see Table 4).

Among the reasons for relinquishment with the greatest difference between groups, Financial Constraints were significant in the lower socioeconomic group, whereas both Health of Human and Life Changes/Family Issues were significant in the higher socioeconomic group.

## 4. Discussion

The purpose of this study was to investigate how a social determinant of health (SDH) proxy, socioeconomic status, is associated with the reasons people relinquish their companion animals. The primary reasons for companion animal relinquishment for those in areas of low socioeconomic status were related to their environment, specifically housing and economic stability or financial constraints, whereas for higher socioeconomic areas, housing was also a factor, but ownership/guardian decisions and human health reasons were more predominant reasons.

In the following sections, the results of this study will be considered in light of the five SDH domains: environment, economic stability, education, healthcare, and social and community. We acknowledge the multifactorial impact of SDH, with each SDH domain overlapping and influencing another domain, but have discussed different factors within single domains.

### 4.1. Social Determinants of Health

#### 4.1.1. Environment

This study identified that housing was the most common reported reason for relinquishing a companion animal. One of the many barriers to securing housing is having a companion animal and the reluctance of landlords or housing management organizations to allow applications from companion animal guardians [11,12,13,14,15,16,17]. For animal guardians trying to exit homelessness, companion animal guardianship is the biggest barrier [18,19,20,21,22]. A study in the United States reported that 42.1% of participants relinquished their companion animal due to moving accommodation or their landlord not allowing companion animals [15]; these numbers were even higher in another study, with 77.5% relinquishing their companion animal due to moving accommodation and 35.1% due to landlord accommodation/contractual conditions [13]. It is reported that between 5% and 25% of people who are unable to secure housing in the United States are companion animal guardians [23]. This challenge affects a large swathe of the population, who have to choose between a roof over their heads or the relationship with their companion animal [24].

We observed that relinquishments reported from lower socioeconomic areas (deciles 1–5) were significantly more likely to be due to housing-related issues than higher socioeconomic (decile 6–10) groups. In addition to reasons such as the landlord not allowing companion animals and moving, other housing-related reasons included environmental challenges, such as not having enough room for the companion animal, no fencing, or being unable to contain the property. The literature also highlights how landlord limitations and the challenge of finding a property with suitable features for animals, such as a yard or fencing, create additional hurdles in obtaining and/or keeping housing with a companion animal [11,12,13,14,15,16,17].

#### 4.1.2. Education

Ownership/guardian decisions emerged as the second most common reason for companion animal relinquishment in our study. Reasons for relinquishment under this category include inadequate selection of the animal, lack of understanding of breed or animal needs, receiving an unwanted gift or acquiring the animal without consent, unwillingness to provide training, or finding the animal’s age no longer suitable. Relinquishments reported from lower socioeconomic areas (deciles 1–5) were more likely due to ownership/ guardian decisions compared to those reported in higher socioeconomic areas (deciles 6–10).

Education is closely associated with literacy, language proficiency, vocational skills, and attainment of higher education [8]. In the context of companion animals, education is fundamental to understanding effective animal training techniques, animal behaviors, and the importance of preventive healthcare measures such as vaccinations [8]. Research conducted on reasons for animal relinquishment across diverse U.S. animal shelters revealed a higher likelihood of relinquishment among guardians with educational levels not exceeding high school [25]. Communities with higher educational levels demonstrated lower rates of stray animal intake in shelters [26]. Furthermore, guardians with lower educational attainment were less likely to have visited a veterinarian within the past 18 months compared to those with a bachelor’s degree (57.1% versus 80.3%, respectively) [27]. This is consistent with the existing literature, which underscores a widespread lack of understanding regarding behavioral and health challenges in companion animals as a prevalent factor leading to relinquishment [25,28,29,30,31,32].

However, access to services and support programs for companion animals appears to be influenced by additional factors, such as barriers related to transportation, availability of veterinary and training services, operational hours, and cost considerations [28], emphasizing the interconnected nature of all domains within the social determinants of health.

#### 4.1.3. Economic Stability

Relinquishing a companion animal due to financial constraints was the third highest reported reason in our study. Alongside employment, debt, and expenses, income is widely regarded as one of the most impactful factors influencing SDH [5]. Companion animal welfare centers on the guardians’ ability to access the necessary resources to fulfill their animals’ cognitive, physical, and environmental needs. This includes their capacity to afford veterinary expenses crucial for ensuring positive welfare outcomes [8].

Relinquishments reported from lower socioeconomic areas (deciles 1–5) were significantly more likely to be due to financial constraints than those reported from higher socioeconomic (decile 6–10) areas. Financial constraint reasons provided by guardians included difficulty affording veterinary fees, including desexing and basic care, with some guardians reporting recent job loss/unemployment and financially associated challenges due to COVID-19. This is supported across the literature, with veterinary care identified as a substantial barrier for guardians to maintain a companion animal relationship [11,13,15,27,33,34,35,36,37,38,39,40,41,42,43,44]. Whilst the literature identified specific financial factors such as grooming costs [45] and expenses for pet food [46,47], our data did not include this level of detail.

#### 4.1.4. Healthcare

Our ‘Health of Human’ relinquishment group included reasons for relinquishment such as allergies, guardian illness, death, or going into care. Here, relinquishment of companion animals was significantly more likely to be reported in higher socioeconomic areas (deciles 6–10) than in lower socioeconomic areas (deciles 1–5). Limited attention has been given to how human health conditions impact animal welfare, but some studies have noted that the health of companion animal guardians can lead to relinquishment [14,31,48,49]. For instance, a Danish study which reviewed 5959 reasons for relinquishment identified that 4453 were owner-related; the primary cause for relinquishment was due to owner health-related conditions (*n* = 1824) of the companion animal guardian [14], consistent with earlier findings [48,50]. We have not been able to identify literature indicating that human health is a common reason for companion animal relinquishment in higher socioeconomic areas, and this is an area that requires further research.

#### 4.1.5. Social and Community

Relationships with other people and support systems for families and communities are drivers of this SDH domain. Life changes and family issues, including challenges such as relationship breakdowns, the arrival of a new baby, and children not getting along with the animal, were reasons why guardians relinquished their companion animals in our study. Relinquishments reported from higher socioeconomic areas (decile 6–10) were more likely to relinquish their companion animal due to life changes/family issues than those reported in lower socioeconomic areas (decile 1–5).

Whilst there is limited literature that identifies time constraints specifically as a challenge to care for a companion animal [14], guardians in the higher socioeconomic group (decile 6–10) were significantly more likely to relinquish a companion animal due to time constraints than those in a lower socioeconomic (decile 1–5) group. The Danish study (*n* = 3204 dogs and *n* = 2755 cats) also found that companion animals were relinquished due to lack of time [14]. In that study, more dogs (14%) than cats (4%) were relinquished for this reason, with the suggestion that this is due to dogs requiring daily exercise and more social interaction than cats [14].

### 4.2. Practical Implications

This study has provided further evidence about the complex interplay between socioeconomic status (SES) and the reasons underlying companion animal relinquishment in Australia. Our findings underscore that companion animal relinquishment rates are notably higher in lower SES areas, with primary reasons revolving around environmental challenges such as housing, educational deficits, and economic instability. These factors reflect broader Social Determinants of Health (SDH) that influence both human and companion animal welfare outcomes. The study reveals a critical need for targeted interventions addressing these disparities to mitigate the impact on vulnerable populations and safeguard the human–animal bond.

Given the multifaceted nature of Social Determinants of Health (SDH) and their impact on reasons for relinquishing companion animals, it is crucial to consider each reason and SDH domain collectively. For instance, if an individual is unable to work due to illness, they may be unable to access veterinary care (Healthcare); financial constraints may further prevent them from affording veterinary services (Economic Stability); lack of transportation to a vet clinic may hinder access (Environment). Moreover, if the individual’s illness prevents them from leaving home, it could lead to inadequate socialization or training for their companion animal, exacerbated by isolation or lack of social visits (Social and Community). Additionally, limited access to suitable parks for dog walking can contribute to negative behaviors (Environment). These interconnected factors underscore the importance of considering all SDH domains when examining reasons for companion animal relinquishment.

In a study in the United States focused on rehoming companion animals from a shelter, 40% of participants indicated that access to free or low-cost veterinary care could have prevented relinquishment [51]. Financial assistance for veterinary expenses is generally limited, often available only in emergency situations [52]. However, over recent years, there has started to be a shift towards providing companion animal support care to underserved communities, or providing more holistic social/community-based veterinary programs [51]. One of the longest running programs of this type is The Humane Society of the United States’ Pets for Life program, which aims to provide a holistic companion animal support service for companion animal guardians and the community [51]. A similar community veterinary service has started to develop within Australia to assist bridging the gap between human health and animal welfare through the development of social enterprise and veterinary clinics together to provide innovative services to the community [53]. Research shows that providing affordable veterinary care to guardians of lower socioeconomic status increases the frequency of veterinary visits for both illness/injury treatment and preventive measures like heartworm prevention and vaccinations [54]. Additionally, in a subsidized grooming program based in New York (*n* = 167), over half of the participating animal guardians reported that grooming costs posed a barrier to maintaining their animals’ welfare [45]. Addressing economic barriers to accessing and providing care for companion animals is crucial for enhancing health and welfare outcomes [51].

### 4.3. Study Limitations and Future Directions

One limitation of this study is the lack of a unified database in Australia specifically dedicated to collecting data on animal relinquishment, coupled with the absence of standardized definitions for relinquishment reasons. As data were sourced from various organizations, each maintaining independent databases, significant efforts were required for data cleaning and harmonization to ensure consistency and meaningful analysis.

Each organization that participated in the study was transparent in providing their data; however, a lot of smaller organizations we contacted did not have data available for the study. As a result, it is likely that this study may underestimate the number of companion animals relinquished. In Australia, we are aware that in higher socioeconomic areas, smaller organizations often operate to assist in rehoming companion animals, and larger shelters/welfare organizations are in areas of lower socioeconomic disadvantage, similar to the situation identified in published research from the USA [55].

This study focused on the relinquishment reason as reported by the animal guardian and as recorded by the organization in the organization database. There might also be some interpretations made by shelter staff when recording the data in the database. A previous study has identified that information provided to shelter staff at the time of relinquishment is limited [56]. Further, there is often more than one reason that a companion animal is relinquished [50], but only one can be recorded, as in our study, which may provide an incomplete picture of animal relinquishment. Our interpretation of the relinquishment reason provided by the organization, and how the reasons have been categorized and linked to the SDH, may vary from others’ interpretations, because SDH domains overlap heavily, and we split them for discussion and analysis purposes. It is also important to note that animal welfare legislation in Australia falls under State and Territory jurisdictions. Given that the data provided for the study comes from several different States, the laws relating to companion animal relinquishment in each jurisdiction may impact on a person’s reason to relinquish their companion animal, such dog registration costs. Furthermore, some organizations may or may not charge a fee for relinquishing a companion animal, and each organization’s policy for accepting companion animals, along with other factors, like whether they are a ‘no kill’ shelter, may also vary. This information was not collected as part of this study, but these factors may also influence a person’s reasons to relinquish their companion animal.’

Finally, data provided are only from a selection of shelters across Australia, and as such, wider geographical generalizations should be made with caution.

## 5. Conclusions

Moving forward, adopting a holistic SDH framework promises to inform targeted policies and interventions aimed at supporting both human and companion animal welfare. By addressing the root causes identified in this study, such as housing insecurity and financial constraints, stakeholders can work towards fostering sustainable solutions that promote the long-term well-being of companion animals and their guardians. This approach contributes to broader societal benefits by enhancing community health and wellbeing.

## Figures and Tables

**Table 1 animals-14-02549-t001:** Total relinquishment data received by organization and data removed due to not meeting study criteria by organization. Total data included in the study *n* = 46,820.

Organization	Total Data Received	Total Data Removed	Total Available for Study
Organization 1	21,591	893	20,698
Organization 2	12,733	975	11,758
Organization 3	8191	2791	5400
Organization 4	13,854	6693	7161
Organization 5	1818	15	1803

**Table 2 animals-14-02549-t002:** Companion animal relinquishment group category with explanation and examples of each category and if human or animal factors.

Relinquishment Group Category	Explanation of Category	Examples	Human/Animal Factor
Animal Health or Behavior	The animal presented with health or behaviors that the guardian was unable to meet	Aggression towards humans or animals Separation anxiety Destructive behavior	Animal
Housing	Guardian is unable to find or maintain suitable accommodation due to animal.	Landlord does not allow pets No fencing Homelessness	Human
Financial Constraints	Guardian is unable to afford animal care needs.	Cannot afford veterinary costs Loss of employment	Human
Time Constraints	Guardian may have demanding jobs or irregular work hours that make it difficult to dedicate time to meet the animal’s welfare needs.	Insufficient time to look after the animal.	Human
Unable to Provide Care	Guardian is unable to meet the animal’s welfare needs due to lack of support or resources to assist them.	Cannot look after animal Unable to rehome	Human
Health of Human	Guardian’s ability to meet the welfare needs of the animal is impaired due to their personal health concerns.	Health of owner Owner going into care Allergies	Human
Life Changes/Family Issues	Circumstances within the guardian’s life have changed since the animal started living with them.	Domestic violence Relationship breakdown/divorce New baby	Human
Ownership/Guardian DecisionsOther	Guardian may have inadvertently chosen a less suitable type of animal for their lifestyle Animals relinquished not by guardian	Grew too big Incompatible activity level Impulse buy	Human

**Table 3 animals-14-02549-t003:** Frequency analysis of all companion animals by species (cats *n* = 27,822/dogs *n* = 18,998) relinquished each year for five financial years (FY 2018/19 to FY 2022/23). The percentage column represents the percentage of total in each year for animals of each species relinquished over the 5 years.

Species	Financial Year (FY)	n	Percentage (%)
Cat	FY 18/19	6681	24.0%
	FY 19/20	6735	24.2%
	FY 20/21	6105	21.9%
	FY 21/22	4796	17.2%
	FY 22/23	3505	12.6%
Dog	FY 18/19	5119	26.9%
	FY 19/20	4687	24.7%
	FY 20/21	3536	18.6%
	FY 21/22	3047	16.0%
	FY 22/23	2609	13.7%

**Table 4 animals-14-02549-t004:** Guardian reasons for relinquishment of companion animals to five Australian animal shelters. Overall association observed of significant associations between categorical variables (socioeconomic status proxy: low/high decile) and reason for relinquishment by socioeconomic group.

Relinquishment Reason	Total (*n* = 46,820)	Low Socioeconomic Status (Decile 1–5)	High Socioeconomic Status (Decile 6–10)	Difference (Low Minus High)	Adjusted Pearson Residual *N*(0, 1)	
					**Decile 1–5**	**Decile 6–10**
Housing	14,622 (31.2%)	9632 (32.2%)	4990 (29.5%)	4642	6.1	−6.1
Ownership/Guardian Decisions	7580 (16.2%)	5160 (17.3%)	2420 (14.3%)	2740	8.3	−8.3
Financial Constraints	5259 (11.2%)	3670 (12.3%)	1589 (9.4%)	2081	9.5	−9.5
Health of Human	4863 (10.4%)	2818 (9.4%)	2045 (12.1%)	773	−9.1	9.1
Unable to Provide Care	3359 (7.2%)	2282 (7.6%)	1077 (6.4%)	1205	5.1	−5.1
Animal Health or Behavior	3039 (6.5%)	1777 (5.9%)	1262 (7.5%)	515	−6.4	6.4
Life Changes/Family Issues	2896 (6.2%)	1609 (5.4%)	1287 (7.6%)	322	−9.6	9.6
Time Constraints	1616 (3.5%)	925 (3.1%)	691 (4.1%)	234	−5.6	5.6
Other	3586 (7.7%)	2026 (6.8%)	1560 (9.2%)	466	−9.5	9.5

Adjusted Pearson residuals of relinquishment reason for both low and high IRSAD deciles. All adjusted residuals *p* < 0.001. Confidence level 95%.

## Data Availability

Restrictions apply to the availability of these data. Data were obtained from third party organizations and are available from the authors with the permission of third party/s.

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
