# Peer review of "Association of Socioeconomic Status and Reasons for Companion Animal Relinquishment"

_animals, 2024, doi:10.3390/ani14172549_

Round 1

Reviewer 1 Report

Comments and Suggestions for Authors

This is a well written manuscript on a timely and important topic. The authors are to be commended on the effort in consolidating an enormous amount of data into a concise discussion. 

I have one comment/recommendation for clarification. On Lines 198-201, some data (Financial Constraints, Health of Human, and Life Changes/Family Issues) changes are referenced as "significantly more than expected" but there is no baseline for what the "expectation" is. Why is an adjusted Pearson Residual >9 "more than expected" but a change >8 (as with Ownership/Guardian Decisions) not "more than expected". Please clarify the reasoning behind the "more than expected" wording in the text. What were the expectations that those values exceeded them? Or change the wording to indicate that they were the reasons with the most significant change and leave out the expectations. 

The rest of the manuscript was very clear and easy to read. This is fascinating work and I hope you continue to build on it. 

Reviewer 2 Report

Comments and Suggestions for Authors

Dear authors, 

I went through this paper with great interests, since understanding what drives dog relinquishment is essential in implementing intervention opportunities to keep companion animals with their owners and inform dog population management and free-roaming dog control intervention strategies.

However, it would be very interesting to also explore potential geographical differences between the legislative provisions regulating dog and cat ownership responsibilities in Australia, and the related relinquishment patterns. To my knowledge, there are no national laws applying to animal welfare in Australia, but all states and territories set and enforce animal welfare standards. Local governments also have legislation relating to the management of companion animals.

In this paper, reference is made to five rescue and shelter organization, but it is not clear if they belong to the same state or territory. In my opinion, this is a weak side of the paper, being the legislation in place, along with the availability and accessibility to services aiming at preventing and mitigating (I.e., permanent and affordable veterinary assistance,  public awareness and education programs, community engagement  on Responsible Pet Ownership, bite prevention and the importance of rabies vaccination) the negative effects of dogs and cats relinquishment. Hence, it would be interesting to know the characteristics of the guardian relinquishment problem with a focus on the existing legislation and the owner’s obligations towards their pets.

The paper deals with both dog and cat relinquishment but the rules to be respected by the owners can be very different. In essence, it would be advisable to include a section describing the local legal background that might have a strong influence on the owner’s decision and concrete possibility to relinquish a companion animal.

As an example, in Italy any person who keeps a companion animal shall be responsible for its health and well-being. According to the Abruzzo Regional Law 147 of 2018, handing over a dog/cat to an animal shelter is deemed to be an act of compliance with this responsibility. In the event that a dog owner or keeper is unable to keep the animal with him for serious reasons, he may ask the Mayor of the Municipality of residence for authorization to deliver the animal to the public municipal long-term shelter. The application must indicate the causes that prevent the dog from being kept. The Mayor shall make a decision within 30 days; in the event of no response the principle of silent consent applies. 

Moreover, surrendering a dog to a shelter may sometimes have a cost anyway, which varies based on several factors such as the owner location, the type of facility, and the age of the pet. Certain facilities charge less per pet for those surrendering entire litters. Some private rescues do not charge a fee if they accept a dog for surrender, but this varies by organization. This and other factors (I.e., if the shelters adopt a “no kill” policy) might influence the owner’s decision (and sometimes real capacity) to surrender his best friend.

Reviewer 3 Report

Comments and Suggestions for Authors

Thank you for a very interesting paper that makes an important contribution to the growing field of veterinary social work, not least in view of the large numbers of records that you have been able to access. I have a few suggestions for you to consider that I feel would improve the paper.

Lines 71-73 It is unclear how poverty and inadequate housing is linked to animals’ aggressive behaviour -could this be expanded upon?

Lines 148-153 I am not sure that the explanation of Adjusted Pearson residuals belongs in the text – perhaps this could be a footnote to Table 4?

Line 220  For animal guardians trying to exit homelessness, companion animal guardianship is the biggest barrier’ – should this be ‘For those trying to exit homelessness, companion animal guardianship is the bigget barrier’?

Lines 287-290 refers to a Danish study but the reference cited is a Canadian study. Also, the (n=1824) is a bit misleading – is this the total number of animals in the study or the number relinquished for owner health-related reasons?

I suggest that referring to ‘lower socioeconomic respondents’ (e.g., line 230, line 244, line 272) in the discussion sections is incorrect, given that you used postcodes to determine IRSADs. Lower IRSAD areas will have inhabitants with higher socioeconomic status and higher IRSAD areas will have inhabitants with lower socioeconomic status. I would suggest that when referring to respondents, you use the terms ‘respondents from lower socioeconomic/higher socioeconomic areas/groups’ throughout.

Lines 290-291 ‘We have not been able to identify literature that links higher socioeconomic human health-related relinquishment of companion animals, and this is an area that requires further research.’ I think there are missing words here that are needed to make the sentence clear – perhaps ‘that links higher socioeconomic groups with human health-related relinquishment…’?

Lines 304-305, reference to ‘the Danish study…. also found’ implies that it has been discussed before but see my comments on lines 287-290
